# The Basic/Helix-Loop-Helix Transcription Factor Family Gene RcbHLH112 Is a Susceptibility Gene in Gray Mould Resistance of Rose (Rosa Chinensis)

**DOI:** 10.3390/ijms242216305

**Published:** 2023-11-14

**Authors:** Chao Ding, Junzhao Gao, Shiya Zhang, Ning Jiang, Dongtao Su, Xinzheng Huang, Zhao Zhang

**Affiliations:** 1Shanxi Center for Testing of Functional Agro-Products, Shanxi Agricultural University, Taiyuan 030031, China; 2Beijing Key Laboratory of Development and Quality Control of Ornamental Crops, Department of Ornamental Horticulture, China Agricultural University, Beijing 100107, China; 17866709530@163.com (J.G.);; 3Department of Entomology, MOA Key Lab of Pest Monitoring and Green Management, College of Plant Protection, China Agricultural University, Beijing 100193, China

**Keywords:** *bHLH*, transcription factor, *Botrytis cinerea*, phylogenetic analysis, expression pattern

## Abstract

The basic/helix–loop–helix (*bHLH*) family is a major family of transcription factors in plants. Although it has been reported that *bHLH* plays a defensive role against pathogen infection in plants, there is no comprehensive study on the *bHLH*-related defence response in rose (*Rosa* sp.). In this study, a genome-wide analysis of *bHLH* family genes (*RcbHLHs*) in rose was carried out, including their phylogenetic relationships, gene structure, chromosome localization and collinearity analysis. Via phylogenetic analysis, a total of 121 *RcbHLH* genes in the rose genome were divided into 21 sub-groups. These *RcbHLHs* are unevenly distributed in all 7 chromosomes of rose. The occurrence of gene duplication events indicates that whole-genome duplication and segmental duplication may play a key role in gene duplication. Ratios of non-synonymous to synonymous mutation frequency (Ka/Ks) analysis showed that the replicated *RcbHLH* genes mainly underwent purification selection, and their functional differentiation was limited. Gene expression analysis showed that 46 *RcbHLHs* were differentially expressed in rose petals upon *B. cinerea* infection. It is speculated that these *RcbHLHs* are candidate genes that regulate the response of rose plants to *B. cinerea* infection. Virus-induced gene silencing (VIGS) confirmed that *RcbHLH112* in rose is a susceptibility factor for infection with *B. cinerea*. This study provides useful information for further study of the functions of the rose *bHLH* gene family.

## 1. Introduction

Transcription factors have been extensively studied in plant growth, development, metabolism and stress response due to their important roles in transcriptional regulation [1]. Transcription factors usually consist of at least DNA-binding domains, transcriptional regulatory domains, oligomerisation sites and nuclear localisation signals [2]. The *bHLH* gene family is one of the most important transcription factor families in plants. Since the discovery of basic/helix–loop–helix (bHLH) motifs [3] with the ability to bind DNA, members of the bHLH protein superfamily have been found to have more and more functions in the basic physiology and development of animals and plants [4,5,6,7,8]. The bHLH domain consists of about 60 amino acids and has two regions with different functions, i.e., the basic domain and the HLH domain. The basic domain is located at the N-terminus of the bHLH domain and acts as a DNA-binding motif. It consists of about 15 amino acids, usually including 6 basic residues. The HLH region contains two amphiphilic alpha helices and a variable-length linker. Two amphiphilic alpha helices of bHLH proteins can interact to form homodimers or heterodimers [9,10]. Some bHLH proteins have been shown to bind to sequences containing a common core element called the E-box (5′-CANNTG-3′). In addition, nucleotides flanking the core elements may also play a role in binding specificity [11].

*bHLH* transcription factors are involved in the regulation of various plant processes, including growth, development and response to biotic and abiotic stresses. The function of *bHLHs* in disease resistance has been characterized in Arabidopsis and many other crops. For example, the wheat *bHLH* transcription factor gene *TabHLH060* increases the susceptibility of transgenic Arabidopsis to *Pseudomonas aeruginosa* [12]. In tomato, *SlybHLH131* increases resistance to yellow leaf curl virus by controlling cell death [13]. Overexpression of jasmonate-responsive *OsbHLH034* in rice results in the induction of bacterial blight resistance via an increase in lignin biosynthesis [14]. In addition, *bHLHs* are also associated with abiotic stress in plants. For example, *MdbHLH130* is the drought response bHLH protein in apple that confers drought tolerance in transgenic tobacco [15]. Overexpression of a *bHLH* gene from *Tamarix hispida* in Arabidopsis can improve salt and drought tolerance by increasing osmotic potential and reducing the accumulation of reactive oxygen species [16]. In Arabidopsis, *bHLH122* is important for drought and osmotic stress resistance and repressing ABA catabolism [17].

Recent research has shown that plant bHLHs can act as a susceptibility gene, negatively regulating plant disease resistance. Zhang and co-authors found that loss of function of the bHLH transcription factor Nrd1 in tomato enhances resistance to *Pseudomonas syringae*. The mutant plants showed increased immunity due to the suppression of a defence gene, Agp1, by Nrd1. This enhanced immunity is independent of the activation of other immunity-associated genes, indicating that Nrd1 plays a specific role in regulating Agp1 expression and susceptibility to Pseudomonas syringae in tomato [18].

Roses (*Rosa* sp.) are commercially the most important ornamental plant, generating tens of billions of dollars in value each year [19]. Grey mould disease of roses caused by *Botrytis cinerea* causes huge losses. There are no reports on the involvement of *bHLH* transcription factors in rose grey mould resistance. To better understand the involvement of the *bHLH* genes in rose resistance against *B. cinerea*, we performed a genome-wide analysis of the *bHLH* family in rose. We further performed RNA-Seq analysis and showed that a large number of genes encoding *bHLH* transcription factors were significantly upregulated upon *B. cinerea* infection, implying that they were involved in the resistance of rose to *B. cinerea* [20]. Importantly, virus-induced gene silencing (VIGS) further confirmed that *RcbHLH112* plays an important role in resistance to *B. cinerea* as a susceptibility gene.

## 2. Results

### 2.1. Identification of RcbHLH Genes in Rose

In the process of identifying *bHLH* family genes in the rose genome, we used the *bHLH* Hidden Markov Model (HMM) file (PF00010) to perform a Hmmsearch search in the rose genome database, and a total of 136 candidate RcbHLH proteins were obtained. MEME (https://meme-suite.org/meme/) (accessed on 11 July 2022) and Pfam database comparison further confirmed that the extracted protein domain was consistent with the characteristics of the family, and finally 121 *RcbHLH* gene members were identified in the rose genome, as these 121 protein sequences had a domain profile consistent with a typical bHLH transcription factor. All *RcbHLH* family genes can be mapped to chromosomes and named *RcbHLH1* to *RcbHLH121* according to their order on chromosomes (Figure 1).

There is a significant difference in the protein size of these *RcbHLHs*. Among the 121 *RcbHLHs*, *RcbHLH25* has the longest amino acid sequence with 1275 amino acids, while the shortest *RcbHLH85* has only 151 amino acids. The average length of RcbHLH protein is 385 aa. The details of all *RcbHLH* genes are listed in Table 1.

### 2.2. Chromosomal Locations, Whole-Genome Duplication and Microsynteny

The 121 *RcbHLH* genes identified are unevenly distributed across 7 rose chromosomes (Figure 1). Chromosome 6 has the most *RcbHLH* genes with 23. There are 20 *RcbHLH* genes on chromosomes 2 and 7. Chromosome 3 has the fewest *RcbHLH* genes, only 9. Meanwhile, 12.37% and 13.22% of *RcbHLH* genes are located in the upper and middle parts of chromosomes 2 and 7, respectively; 9. 92% of the *RcbHLH* genes are located in the upper and middle parts of chromosome 5; 9.09% and 10.74% of the genes are distributed in the middle and lower parts of chromosomes 1 and 4, respectively; and 19.01% of the *RcbHLH* genes are distributed on chromosome 6.

Tandem and segmental duplication play an important role in the expansion of gene families and the generation of new gene functions. On further examination of the repetitive events, we found that there were 16 gene pairs in this family, all of which were whole-genome duplication (WGD) or segmental duplication, while there were gene pairs on different chromosomes, indicating that these genes were paralogous genes. The microsynteny of these *RcbHLH* genes is shown in Figure 2.

To investigate the selective constraints between duplicated *RcbHLH* genes, the ratios of non-synonymous mutation frequency (Ka) to synonymous mutation frequency (Ks) of 16 gene pairs were calculated (Table 2). In general, Ka/Ks > 1 is consistent with positive selection, whereas Ka/Ks < 1 indicates purifying selection. The Ka/Ks ratio of all 16 repetitive gene pairs is less than 1 (Table 2), indicating limited functionally divergent purifying selection during the evolutionary history of the repetitive *RcbHLH* genes.

### 2.3. Phylogenetic and Exon-Intron Structural Analysis of Rose bHLH Genes

We used the neighbour-joining method (NJ) method to reconstruct the phylogeny of all *RcbHLH* genes and constructed a phylogenetic tree. The results of the follow-up analysis of the exon–intron structure are consistent with those of the phylogenetic analysis (Figure 3). Most genes clustered in the same group have similar genetic structures, especially in terms of the number of introns, such as *RcbHLH10*, *RcbHLH26* and *RcbHLH120*. However, there were some exceptions. For example, *RcbHLH58* and *RcbHLH105* contain different numbers of introns. In addition, their intron length is very variable, ranging from tens to thousands of nucleotides. These results indicate that there is a highly conservative structure in the *RcbHLH* subfamily and that there is sequence diversity between different *RcbHLH* groups.

In addition, there is increasing evidence that *bHLH* transcription factors play a key role in disease resistance in various plant species (Table 3). To assess the evolutionary relationship between *RcbHLHs* and *AtbHLHs* genes, we constructed a composite phylogenetic tree (Figure 4). According to The Arabidopsis Information Resources (TAIR) (http://www.arabidopsis.org/) (accessed on 11 July 2022), there are 158 *AtbHLH* genes in Arabidopsis. These members can be divided into 21 different groups. The results confirmed the previously proposed classification of the *bHLH* family. The subfamily VIII (a+b+c) contains 31 proteins, and the subfamily IVb contains 3 proteins. The bootstrap values of some branches in the phylogenetic tree are low, which may be due to the short bHLH domain and relatively little information other than highly conserved information.

### 2.4. Expression of RcbHLH Genes in Response to B. cinerea Infection

A growing body of evidence from different plant species indicates that plant *bHLH* transcription factors play an important role in pathogen response. To investigate the role of *bHLH* genes in *B. cinerea* resistance in rose, we analysed transcriptome data from rose petals inoculated with the pathogen at 30 and 48 hpi. The 30 hpi time point represents the early response to infection, whereas the 48 hpi time point corresponds to the late response (Table 4). The log_2_Ratio transformed expression profiles were obtained from the RNA-seq dataset [20]. A total of 21 *RcbHLH* genes (*RcbHLH17*, *RcbHLH21*, *RcbHLH29*, *RcbHLH34*, *RcbHLH40*, *RcbHLH44*, *RcbHLH46*, *RcbHLH59*, *RcbHLH62*, *RcbHLH67*, *RcbHLH72*, *RcbHLH75*, *RcbHLH80*, *RcbHLH90*, *RcbHLH99*, *RcbHLH101*, *RcbHLH106*, *RcbHLH108*, *RcbHLH111*, *RcbHLH112* and *RcbHLH115*) were upregulated, suggesting that they may be the key regulators of *B. cinerea* infection and influence the disease resistance of rose. To further verify the expression profile of RNA-seq, the expression of 4 *RcbHLHs* was analysed by RT-qPCR. The results of the RT-qPCR analysis were consistent with those of the transcriptome analysis (Figure 5).

### 2.5. RcbHLH112 Is a Susceptibility Gene to B. cinerea in Rose

To further investigate the potential role of *B. cinerea*-induced *RcbHLH* genes in pathogen resistance, we used VIGS to knock down the expression of *RcbHLH112* in rose petals. The reason for selecting *RcbHLH112* for this VIGS study was that *RcbHLH112* is one of the most upregulated *RcbHLHs* after *B. cinerea* infection (Table 4). To silence *RcbHLH112* in rose petals, we cloned the 256 bp fragment of *RcbHLH112* into the tobacco rattle virus (*TRV2*) vector to generate *TRV-RcbHLH112*. Agrobacterium tumefaciens carrying *TRV-RcbHLH112* and *TRV1* were co-infiltrated into rose petals to produce rose petals with *RcbHLH112* silencing. The infiltrated rose petals were then inoculated with *B. cinerea*. Compared with the control petals (*TRV-GFP*) inoculated with *TRV* with a GFP sequence, petals inoculated with *TRV-RcbHLH112* showed attenuation of disease symptoms, with a significant reduction in the size of the lesion (Figure 6A,B). In addition, we used RT-qPCR to verify the silencing efficiency of VIGS (Figure 6C). These results show that *RcbHLH112* is a susceptibility factor for rose resistance against *B. cinerea* and that its silencing increases resistance to *B. cinerea* in rose.

## 3. Discussion

The *bHLH* genes play important roles in plant growth, development and defence. In this study, we comprehensively analysed the *RcbHLH* family, including phylogeny, gene structure, chromosome localization, gene duplication events, sequence characteristics and expression profile analysis. We demonstrated that *RcbHLH112* is involved in the regulation of resistance to *B. cinerea* in rose.

It was found that the number of *RcbHLH* genes in rose (121) was lower than that in Arabidopsis (158), rice (167), potato (124) and maize (208) [26,27,28], indicating that the *bHLH* gene has expanded to different degrees in different plants. Gene replication plays a very important role in gene family expansion. In this study, 16 replication events were identified in 56 *RcbHLHs*, all of which involved segmental duplication. The Ka/Ks ratio of the 16 *RcbHLH* repeats indicates that the *RcbHLH* gene family is under purifying selection, suggesting a highly conserved evolution. The phylogenetic relationship of *bHLH* between rose and Arabidopsis showed that most evolutionary branches contained different numbers of AtbHLH and RcbHLH proteins, indicating that the two species showed conservative evolution. These results suggest that the species-specific *bHLH* gene was lost in rose or gained in the Arabidopsis phylogeny after divergence from the most recent common ancestor.

The role of *RcbHLH* in *B. cinerea* resistance is still unclear. In this study, we constructed a phylogenetic tree of known resistance-related *bHLHs* and found that the *bHLHs* involved in disease resistance were distributed in groups Ia, Ib, IVb, IVc and Ⅲ(d+e+f). According to the expression in response to *B. cinerea* infection, we identified 21 *RcbHLHs* that could be involved in *B. cinerea* resistance in rose petals. Interestingly, most of the *RcbHLH* genes induced by *B. cinerea* are associated with segmental duplication events. The *RcbHLH112* belonging to Ib was on the same evolutionary branch as the *B. cinerea* resistance-related *bHLH* found in many different species and was significantly induced by *B. cinerea* at 30 hpi and 48 hpi. Therefore, *RcbHLH112* should be considered as an important candidate gene involved in the regulation of disease resistance in rose. The results of VIGS in rose petals showed that silencing of *RcbHLH112* improved resistance to *B. cinerea*, indicating that it is a susceptibility factor of rose in *B. cinerea* infection process.

## 4. Materials and Methods

### 4.1. Identification and Characteristics of the bHLH Genes in Rose Genome

The complete genome data were downloaded from the Rosa chinensis ‘Old Blush’ genome website https://lipm-browsers.toulouse.inra.fr/pub/RchiOBHm-V2/ (accessed on 5 July 2022) for local alignment and analysis. To identify the non-redundant *bHLH* genes in the rose genome, first, the common protein sequence of the *bHLH* Hidden Markov Model (HMM) (PF00010) was downloaded from the Pfam website (http://pfam.xfam.org) (accessed on 11 July 2022). Then, using the HMM profile as a query, the rose genome was searched using the hmmblast function and all sequences were identified as containing bHLH domains with an E-value of <1 × 10^−3^ in rose. Finally, the protein and DNA sequences of the above rose *bHLH* members were extracted using the TBTools tool, and all candidate *RcbHLHs* were verified using the functional structure identified by MEME (https://meme-suite.org/meme/) (accessed on 11 July 2022) and the Pfam database to determine the final family members.

### 4.2. Mapping bHLH Genes on Rose Chromosomes

The physical locations of 121 genes were extracted from the genomic gff3 annotation file of rose. Mapchart 2.2 software was used to visualise the distribution of *bHLH* genes on 7 rose chromosomes [29].

### 4.3. Phylogenetic Analyses and Structure Analysis

A total of 158 Arabidopsis bHLH protein sequences were collected from TAIR (http://www.arabidopsis.org/) (accessed on 11 July 2022). The bHLH protein sequences of Arabidopsis thaliana and rose were compared using ClustalW. The bHLH sequence alignments were used for phylogenetic analysis. The phylogenetic tree was constructed using MEGA6 software, calculating the advance distance via p-distance, estimating the amino acid substitution at each site, performing 1000 bootstrap sampling steps and constructing the phylogenetic tree via the NJ method [30]. The gene structure map and functional structure map of *RcbHLH* were completed using TBtools [31].

### 4.4. Collinearity Analyses and Calculation of Ratios of Non-Synonymous (Ka) to Synonymous (Ks) Nucleotide Substitution

We used TBtools to analyse the collinearity of *bHLH* members and calculate the ratio of Ka/Ks [32].

### 4.5. Expression of RcbHLHs in Response to B. cinerea

The RNA-Seq data of rose petals infected with *B. cinerea* can be obtained from the National Center for Biotechnology Information (NCBI) database, accession number PRJNA414570. Clean sequencing reads were mapped to the rose reference genome. Reads per kb per million reads (RPKM) were used to obtain gene expression level. The gene expression level of *RcbHLH* was calculated as reads per kb per million reads. Differential expression analysis based on Log2 fold change was analysed using DEseq2. To verify the results of RNA-Seq, quantitative PCR (qPCR) was used to analyse the expression of 4 *RcbHLH* genes. Therefore, total RNA was extracted from rose petals 30 and 48 h after inoculation using the hot borate method [33]. First-strand cDNA was synthesised using HiScript II Q Select RT SuperMix (Vazyme) in 20 μL reaction volume, and 1 μg DNase-treated RNA was used. SYBR Green Master Mix (Takara, Dalian, China) was used for the qPCR reaction, and detection was performed on a StepOnePlus real-time PCR system (Thermo Fisher Scientific, Waltham, MA, USA). RcUBI2 was used as an internal control. Expression was analysed via the delta–delta–CT method. All primers used for qPCR are listed in Table 5.

### 4.6. VIGS and B. cinerea Inoculation Assays

To generate the *TRV-RcbHLH112* constructor, the 256 bp fragment of *RcbHLH112* was amplified using a pair of primers, *RcbHLH112-F*(5’-GGGGGACAAGTTTGTACAAAAAAGCAGGCTTCTGAGGAAGAAGGAGCCGAAG-3’) and *RcbHLH112-R*(5’-GGGGGACCACTTTGTACAAGAAAGCTGGGTCCTCAGCTTAGCCTTGTGGAGT-3’).

The VIGS process involved taking individual petals from the outermost whorls of the rose at the second stage of flowering. A 15 mm disc was then cut from the centre of each petal. Agrobacterium tumefaciens cultures containing constructs expressing *TRV1* and *TRV2* were mixed 1:1 and infiltrated into the petal disc under vacuum [34]. On day 6 after infection, the petal disc was inoculated with *B. cinerea*. A minimum of 48 petal discs were used for genes, and VIGS was repeated at least three times. After inoculation with *B. cinerea*, Student’s *t*-test was performed to determine the significance of lesion size.

## 5. Conclusions

In this study, a genome-wide analysis of the *RcbHLH* family genes was performed, including phylogenetic relationship, collinearity and expression analysis. A total of 121 non-redundant *bHLH* family members were identified. These *RcbHLH* family genes were classified into 21 groups based on phylogeny and conserved domains. Expression analysis showed that the transcriptional regulation of some *RcbHLH* family genes was induced by *B. cinerea* infection in rose petals. Furthermore, plant *bHLHs* involved in disease resistance tended to cluster on the same branch of the phylogenetic tree. Based on these analyses, we used VIGS to further demonstrate that *RcbHLH112* is a susceptibility factor of rose in *B. cinerea* infection process. The information provided by these results can promote further functional analysis of the *RcbHLH* gene in rose.

## Figures and Tables

**Figure 1 ijms-24-16305-f001:**
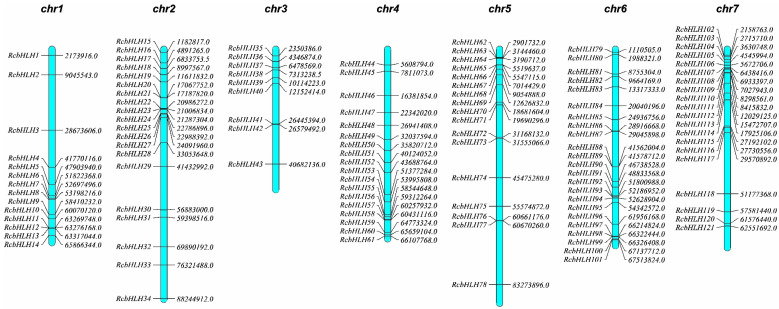
Chromosome localization of rose *bHLH* family members. The physical distribution of each *RcbHLH* gene is listed on the seven chromosomes of *Rose chinensis*.

**Figure 2 ijms-24-16305-f002:**
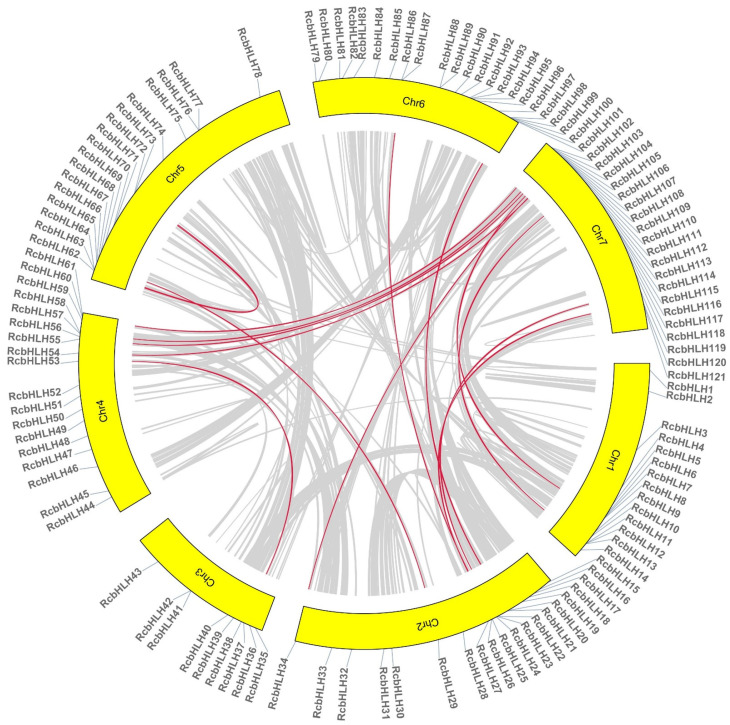
Microsyntenic analyses of the rose *bHLH* transcription factors in the *Rose chinensis* genome. Circular visualization of rose *bHLH* transcription factors is mapped onto different chromosomes using Circos [21]. The red lines indicate rose *bHLH* genes with a syntenic relationship. The grey lines represent all syntenic blocks in the genome of *Rose chinensis*.

**Figure 3 ijms-24-16305-f003:**
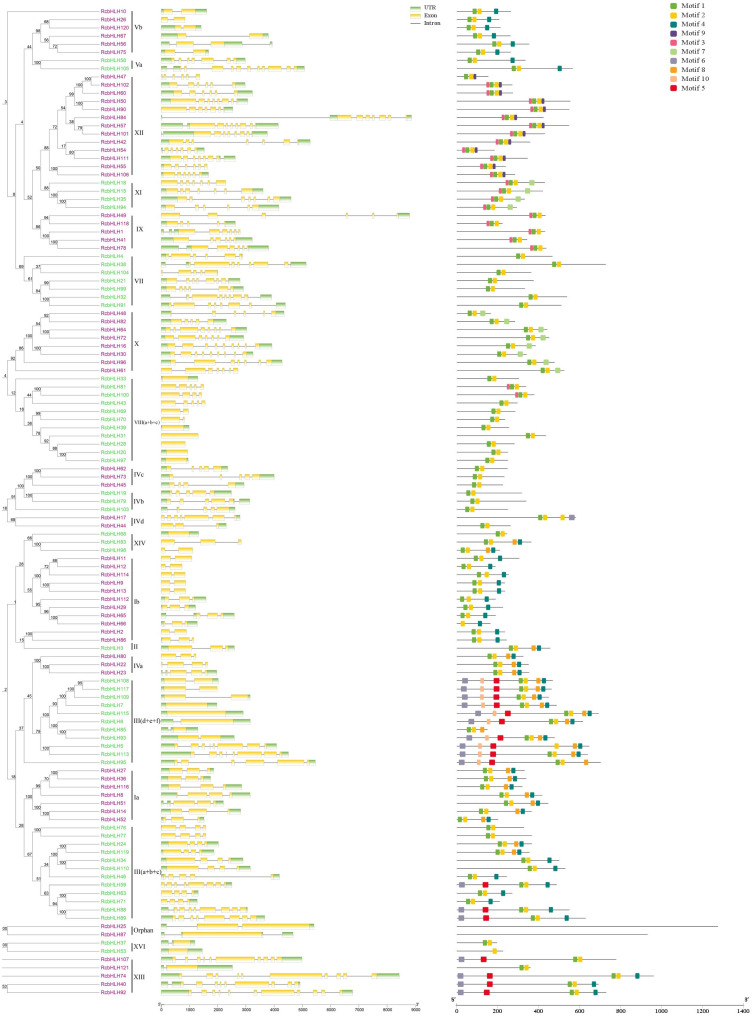
Phylogenetic analyses, DNA structures and protein motifs of the *bHLH* gene family in rose. Complete alignments of all rose *bHLH* proteins were used to construct a phylogenetic tree using the neighbour-joining method. The left represents gene structures. The green boxes, yellow boxes and grey lines in the exon–intron structure diagram represent UTRs, exons and introns, respectively. The right represents protein motifs in the *bHLH* members. The colourful boxes delineate different motifs (unit: aa). The scale on the bottom is provided as a reference.

**Figure 4 ijms-24-16305-f004:**
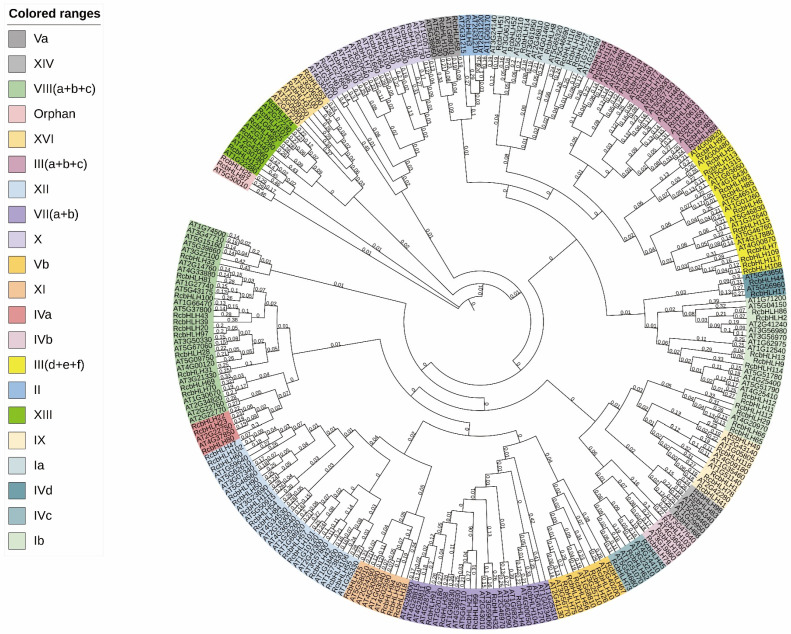
Phylogenetic analyses of the rose *bHLH* transcription factors. Composite phylogenetic tree of rose and Arabidopsis *bHLH* transcription factors. The bootstrap values are indicated on the nodes of the branches.

**Figure 5 ijms-24-16305-f005:**
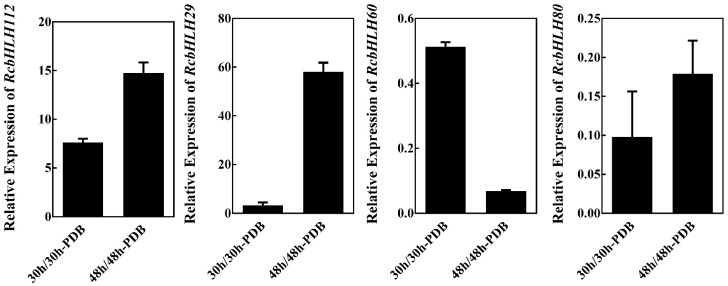
Validation of RNA-Seq results using qRT-PCR. RhUbi was used as an internal control. Expression profile data of four *RcbHLH* genes at 30 hpi and 48 hpi after *B. cinerea* inoculation were obtained using qRT-PCR. Values are the means of three replicates ± SD. The primers used are listed in Table 5.

**Figure 6 ijms-24-16305-f006:**
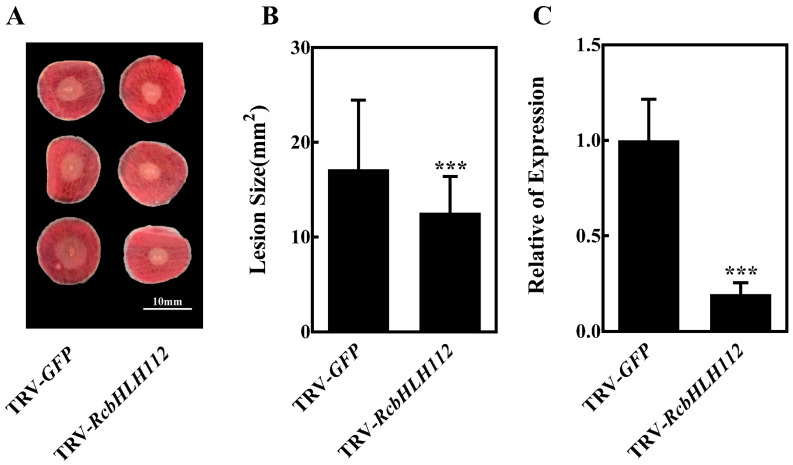
Functional analysis of rose *bHLH* transcription factor gene *RcbHLH112*. (**A**) Compromised *B. cinerea* resistance upon silencing of *RcbHLH112* at 60 hpi post inoculation. A recombinant tobacco rattle virus (*TRV*) targeting *RcbHLH112* (*TRV-RcbHLH112*) was used for the gene silencing, and a *TRV* with GFP sequence (*TRV-GFP*) was used as the control. (**B**) Quantification of *B. cinerea* disease lesions on *TRV-RcbHLH112-* and *TRV-GFP*-inoculated rose petal discs. The graph shows the lesion size from three biological replicates (*n* = 48) with the standard deviation. (**C**) Expression of *RcbHLH112* relative to that in the control at 6 days post silencing, before the infection with *B. cinerea* (0 hpi). All statistical analyses were performed using Student’s *t*-test; *** *p* < 0.001.

**Table 1 ijms-24-16305-t001:** Members of the *RcbHLH* gene family as predicted in *R. chinensis* genome sequence.

Gene	Accession Number ^1^	Chr. ^2^	Position ^3^	Intron	Extron	CDS (bp)	Amino Acids	Clade
*RcbHLH1*	RchiOBHm_Chr1g0314891	1	2,173,916	7	8	1296	432	Ⅸ
*RcbHLH2*	RchiOBHm_Chr1g0321521	1	9,045,543	1	2	708	236	Ⅰb
*RcbHLH3*	RchiOBHm_Chr1g0337011	1	28,673,606	2	3	1371	457	Ⅱ
*RcbHLH4*	RchiOBHm_Chr1g0348781	1	41,770,117	5	6	1404	468	Ⅶ
*RcbHLH5*	RchiOBHm_Chr1g0355211	1	47,903,940	7	8	1944	648	Ⅲ(d+e+f)
*RcbHLH6*	RchiOBHm_Chr1g0360001	1	51,822,366	1	2	1851	617	Ⅲ(d+e+f)
*RcbHLH7*	RchiOBHm_Chr1g0360811	1	52,697,497	0	1	1464	488	Ⅲ(d+e+f)
*RcbHLH8*	RchiOBHm_Chr1g0361191	1	53,198,215	3	4	1254	418	Ⅰa
*RcbHLH9*	RchiOBHm_Chr1g0368211	1	58,410,232	1	2	708	236	Ⅰb
*RcbHLH10*	RchiOBHm_Chr1g0370561	1	60,070,116	1	2	795	265	Ⅴb
*RcbHLH11*	RchiOBHm_Chr1g0376001	1	63,269,742	1	2	918	306	Ⅰb
*RcbHLH12*	RchiOBHm_Chr1g0376011	1	63,276,165	1	2	570	190	Ⅰb
*RcbHLH13*	RchiOBHm_Chr1g0376061	1	63,317,042	1	2	711	237	Ⅰb
*RcbHLH14*	RchiOBHm_Chr1g0380101	1	65,866,341	2	3	1098	366	Ⅰa
*RcbHLH15*	RchiOBHm_Chr2g0085911	2	1,182,817	6	7	1263	421	Ⅺ
*RcbHLH16*	RchiOBHm_Chr2g0091241	2	4,891,265	6	7	1158	386	Ⅹ
*RcbHLH17*	RchiOBHm_Chr2g0093571	2	6,833,754	6	7	1743	581	Ⅳd
*RcbHLH18*	RchiOBHm_Chr2g0096091	2	8,997,567	6	7	1293	431	Ⅺ
*RcbHLH19*	RchiOBHm_Chr2g0099391	2	11,611,832	4	5	954	318	Ⅳb
*RcbHLH20*	RchiOBHm_Chr2g0105811	2	17,067,752	0	1	753	251	Ⅷ(a+b+c)
*RcbHLH21*	RchiOBHm_Chr2g0105931	2	17,187,821	7	8	1128	376	Ⅶ
*RcbHLH22*	RchiOBHm_Chr2g0109611	2	20,986,271	3	4	1056	352	Ⅳa
*RcbHLH23*	RchiOBHm_Chr2g0109621	2	21,006,834	4	5	1062	354	Ⅳa
*RcbHLH24*	RchiOBHm_Chr2g0109941	2	21,287,305	3	4	1101	367	Ⅲ(a+b+c)
*RcbHLH25*	RchiOBHm_Chr2g0111201	2	22,786,895	2	3	3825	1275	Orphan
*RcbHLH26*	RchiOBHm_Chr2g0111351	2	22,988,391	1	2	624	208	Ⅴb
*RcbHLH27*	RchiOBHm_Chr2g0112221	2	24,091,959	2	3	996	332	Ⅰa
*RcbHLH28*	RchiOBHm_Chr2g0120331	2	33,053,648	0	1	849	283	Ⅷ(a+b+c)
*RcbHLH29*	RchiOBHm_Chr2g0126861	2	41,432,994	2	3	678	226	Ⅰb
*RcbHLH30*	RchiOBHm_Chr2g0139261	2	56,883,003	8	9	1026	342	Ⅹ
*RcbHLH31*	RchiOBHm_Chr2g0141851	2	59,398,516	0	1	1308	436	Ⅷ(a+b+c)
*RcbHLH32*	RchiOBHm_Chr2g0152511	2	69,890,195	8	9	1617	539	Ⅶ
*RcbHLH33*	RchiOBHm_Chr2g0160481	2	76,321,485	0	1	912	304	Ⅷ(a+b+c)
*RcbHLH34*	RchiOBHm_Chr2g0176421	2	88,244,910	3	4	1503	501	Ⅲ(a+b+c)
*RcbHLH35*	RchiOBHm_Chr3g0451111	3	2,350,386	6	7	999	333	Ⅺ
*RcbHLH36*	RchiOBHm_Chr3g0454211	3	4,346,874	2	3	1020	340	Ⅰa
*RcbHLH37*	RchiOBHm_Chr3g0457291	3	6,478,569	1	2	591	197	ⅩⅥ
*RcbHLH38*	RchiOBHm_Chr3g0458701	3	7,313,238	8	9	2184	728	Ⅶ
*RcbHLH39*	RchiOBHm_Chr3g0462431	3	10,114,223	0	1	768	256	Ⅷ(a+b+c)
*RcbHLH40*	RchiOBHm_Chr3g0465361	3	12,152,414	9	10	2079	693	ⅩⅢ
*RcbHLH41*	RchiOBHm_Chr3g0480621	3	26,445,394	5	6	1032	344	Ⅸ
*RcbHLH42*	RchiOBHm_Chr3g0480751	3	26,579,491	6	7	1077	359	Ⅻ
*RcbHLH43*	RchiOBHm_Chr3g0493491	3	40,682,138	4	5	891	297	Ⅷ(a+b+c)
*RcbHLH44*	RchiOBHm_Chr4g0390311	4	5,608,794	2	3	792	264	Ⅳd
*RcbHLH45*	RchiOBHm_Chr4g0392401	4	7,811,073	3	4	678	226	Ⅳa
*RcbHLH46*	RchiOBHm_Chr4g0399211	4	16,381,854	4	5	735	245	Ⅲ(a+b+c)
*RcbHLH47*	RchiOBHm_Chr4g0403251	4	22,342,021	5	6	465	155	Ⅻ
*RcbHLH48*	RchiOBHm_Chr4g0405961	4	26,941,409	5	6	501	167	Ⅹ
*RcbHLH49*	RchiOBHm_Chr4g0409001	4	32,037,594	5	6	1302	434	Ⅸ
*RcbHLH50*	RchiOBHm_Chr4g0412071	4	35,820,711	7	8	1662	554	Ⅻ
*RcbHLH51*	RchiOBHm_Chr4g0415421	4	40,124,051	4	5	1341	447	Ⅰa
*RcbHLH52*	RchiOBHm_Chr4g0418301	4	43,688,763	2	3	609	203	Ⅰa
*RcbHLH53*	RchiOBHm_Chr4g0425781	4	51,377,284	0	1	681	227	ⅩⅥ
*RcbHLH54*	RchiOBHm_Chr4g0429161	4	53,995,807	6	7	558	186	Ⅻ
*RcbHLH55*	RchiOBHm_Chr4g0434901	4	58,544,648	5	6	720	240	Ⅻ
*RcbHLH56*	RchiOBHm_Chr4g0435901	4	59,312,260	3	4	1062	354	Ⅴb
*RcbHLH57*	RchiOBHm_Chr4g0437041	4	60,257,934	8	9	1647	549	Ⅻ
*RcbHLH58*	RchiOBHm_Chr4g0437281	4	60,431,122	6	7	1008	336	Ⅴa
*RcbHLH59*	RchiOBHm_Chr4g0443741	4	64,773,328	7	8	1464	488	Ⅲ(a+b+c)
*RcbHLH60*	RchiOBHm_Chr4g0445091	4	65,659,106	5	6	825	275	Ⅻ
*RcbHLH61*	RchiOBHm_Chr4g0445691	4	66,107,770	6	7	1578	526	Ⅹ
*RcbHLH62*	RchiOBHm_Chr5g0004471	5	2,901,732	4	5	747	249	Ⅳa
*RcbHLH63*	RchiOBHm_Chr5g0004791	5	3,144,460	3	4	816	272	Ⅲ(a+b+c)
*RcbHLH64*	RchiOBHm_Chr5g0004831	5	3,190,712	7	8	1329	443	Ⅹ
*RcbHLH65*	RchiOBHm_Chr5g0008581	5	5,519,637	3	4	573	191	Ⅰb
*RcbHLH66*	RchiOBHm_Chr5g0008601	5	5,547,115	2	3	495	165	Ⅰb
*RcbHLH67*	RchiOBHm_Chr5g0010631	5	7,014,429	1	2	789	263	Ⅴb
*RcbHLH68*	RchiOBHm_Chr5g0013411	5	9,054,888	0	1	741	247	ⅩⅣ
*RcbHLH69*	RchiOBHm_Chr5g0018101	5	12,626,832	1	2	861	287	Ⅷ(a+b+c)
*RcbHLH70*	RchiOBHm_Chr5g0024601	5	18,681,604	1	2	711	237	Ⅷ(a+b+c)
*RcbHLH71*	RchiOBHm_Chr5g0025741	5	19,690,297	3	4	633	211	Ⅲ(a+b+c)
*RcbHLH72*	RchiOBHm_Chr5g0036871	5	31,168,132	6	7	1356	452	Ⅹ
*RcbHLH73*	RchiOBHm_Chr5g0037201	5	31,555,063	4	5	699	233	Ⅳa
*RcbHLH74*	RchiOBHm_Chr5g0048491	5	45,475,282	9	10	2886	962	ⅩⅢ
*RcbHLH75*	RchiOBHm_Chr5g0053301	5	55,574,872	1	2	792	264	Ⅴb
*RcbHLH76*	RchiOBHm_Chr5g0056871	5	60,661,176	3	4	987	329	Ⅲ(a+b+c)
*RcbHLH77*	RchiOBHm_Chr5g0056881	5	60,670,256	3	4	1101	367	Ⅲ(a+b+c)
*RcbHLH78*	RchiOBHm_Chr5g0077341	5	83,273,897	6	7	1314	438	Ⅸ
*RcbHLH79*	RchiOBHm_Chr6g0245181	6	1,110,505	5	6	1020	340	Ⅳb
*RcbHLH80*	RchiOBHm_Chr6g0246251	6	1,988,321	2	3	975	325	Ⅳa
*RcbHLH81*	RchiOBHm_Chr6g0253641	6	8,755,304	4	5	1017	339	Ⅷ(a+b+c)
*RcbHLH82*	RchiOBHm_Chr6g0254731	6	9,664,169	4	5	855	285	Ⅹ
*RcbHLH83*	RchiOBHm_Chr6g0257881	6	13,317,333	2	3	1095	365	ⅩⅣ
*RcbHLH84*	RchiOBHm_Chr6g0264701	6	20,040,197	7	8	1272	424	Ⅻ
*RcbHLH85*	RchiOBHm_Chr6g0268091	6	24,936,757	1	2	453	151	Ⅲ(d+e+f)
*RcbHLH86*	RchiOBHm_Chr6g0270891	6	28,916,670	2	3	732	244	Ⅰb
*RcbHLH87*	RchiOBHm_Chr6g0271001	6	29,045,898	2	3	2796	932	Orphan
*RcbHLH88*	RchiOBHm_Chr6g0278441	6	41,562,004	8	9	1653	551	Ⅲ(a+b+c)
*RcbHLH89*	RchiOBHm_Chr6g0278471	6	41,578,713	7	8	1890	630	Ⅲ(a+b+c)
*RcbHLH90*	RchiOBHm_Chr6g0283511	6	46,738,527	6	7	1650	550	Ⅻ
*RcbHLH91*	RchiOBHm_Chr6g0285491	6	48,833,566	7	8	1533	511	Ⅶ
*RcbHLH92*	RchiOBHm_Chr6g0288541	6	51,800,989	10	11	2190	730	ⅩⅢ
*RcbHLH93*	RchiOBHm_Chr6g0288981	6	52,186,951	1	2	1434	478	Ⅲ(d+e+f)
*RcbHLH94*	RchiOBHm_Chr6g0289601	6	52,628,904	5	6	882	294	Ⅺ
*RcbHLH95*	RchiOBHm_Chr6g0291161	6	54,342,571	5	6	2109	703	Ⅲ(d+e+f)
*RcbHLH96*	RchiOBHm_Chr6g0301601	6	61,956,169	6	7	1434	478	Ⅹ
*RcbHLH97*	RchiOBHm_Chr6g0308101	6	66,214,825	0	1	750	250	Ⅷ(a+b+c)
*RcbHLH98*	RchiOBHm_Chr6g0308241	6	66,322,449	1	2	633	211	ⅩⅣ
*RcbHLH99*	RchiOBHm_Chr6g0308251	6	66,326,408	5	6	1002	334	Ⅶ
*RcbHLH100*	RchiOBHm_Chr6g0309431	6	67,137,708	3	4	1137	379	Ⅷ(a+b+c)
*RcbHLH101*	RchiOBHm_Chr6g0310101	6	67,513,823	8	9	1293	431	Ⅻ
*RcbHLH102*	RchiOBHm_Chr7g0180121	7	2,158,763	5	6	816	272	Ⅻ
*RcbHLH103*	RchiOBHm_Chr7g0181001	7	2,715,710	4	5	750	250	Ⅳb
*RcbHLH104*	RchiOBHm_Chr7g0182341	7	3,630,748	4	5	1092	364	Ⅶ
*RcbHLH105*	RchiOBHm_Chr7g0183781	7	4,545,994	11	12	1701	567	Ⅴa
*RcbHLH106*	RchiOBHm_Chr7g0185551	7	5,672,706	5	6	855	285	Ⅻ
*RcbHLH107*	RchiOBHm_Chr7g0186541	7	6,438,416	9	10	2337	779	ⅩⅢ
*RcbHLH108*	RchiOBHm_Chr7g0187141	7	6,933,397	1	2	1407	469	Ⅲ(d+e+f)
*RcbHLH109*	RchiOBHm_Chr7g0187261	7	7,027,943	1	2	1350	450	Ⅲ(d+e+f)
*RcbHLH110*	RchiOBHm_Chr7g0188921	7	8,298,561	3	4	1593	531	Ⅲ(a+b+c)
*RcbHLH111*	RchiOBHm_Chr7g0189021	7	8,415,832	7	8	1041	347	Ⅻ
*RcbHLH112*	RchiOBHm_Chr7g0193761	7	12,029,125	2	3	573	191	Ⅰb
*RcbHLH113*	RchiOBHm_Chr7g0197531	7	15,472,708	7	8	1920	640	Ⅲ(d+e+f)
*RcbHLH114*	RchiOBHm_Chr7g0199961	7	17,925,106	1	2	765	255	Ⅰb
*RcbHLH115*	RchiOBHm_Chr7g0209751	7	27,192,104	0	1	2082	694	Ⅲ(d+e+f)
*RcbHLH116*	RchiOBHm_Chr7g0210101	7	27,730,558	2	3	963	321	Ⅰa
*RcbHLH117*	RchiOBHm_Chr7g0212241	7	29,570,891	1	2	1392	464	Ⅲ(d+e+f)
*RcbHLH118*	RchiOBHm_Chr7g0227911	7	51,177,367	4	5	672	224	Ⅸ
*RcbHLH119*	RchiOBHm_Chr7g0233161	7	57,581,440	3	4	1068	356	Ⅲ(a+b+c)
*RcbHLH120*	RchiOBHm_Chr7g0236841	7	61,576,445	1	2	645	215	Ⅴb
*RcbHLH121*	RchiOBHm_Chr7g0237511	7	62,551,696	1	2	1083	361	ⅩⅢ

^1^ Available at https://lipm-browsers.toulouse.inra.fr/pub/RchiOBHm-V2/ (accessed on 5 July 2022 ). ^2^ Chromosome. ^3^ Starting position (b).

**Table 2 ijms-24-16305-t002:** Duplication analysis of the *RcbHLH* gene family.

Sequence 1	Sequence 2	Ka	Ks	Ka/Ks	Effective Len	Average S-Sites	Average N-Sites
*RcbHLH7*	*RcbHLH109*	0.443081	NaN	NaN	1275	291.6667	983.3333
*RcbHLH11*	*RcbHLH114*	0.462303	NaN	NaN	597	133.8333	463.1667
*RcbHLH20*	*RcbHLH97*	0.328431	2.687639	0.1222	636	140	496
*RcbHLH21*	*RcbHLH99*	0.48712	1.774278	0.274545	885	202.0833	682.9167
*RcbHLH24*	*RcbHLH119*	0.287838	1.620301	0.177645	1032	222.1667	809.8333
*RcbHLH25*	*RcbHLH87*	0.457511	2.840734	0.161054	2733	555.5833	2177.417
*RcbHLH26*	*RcbHLH120*	0.59898	1.798421	0.333059	555	133.8333	421.1667
*RcbHLH29*	*RcbHLH65*	0.455457	4.125944	0.110389	537	122.5	414.5
*RcbHLH34*	*RcbHLH110*	0.370849	2.185043	0.169722	1443	325.9167	1117.083
*RcbHLH37*	*RcbHLH53*	0.392481	1.639823	0.239344	588	153.5833	434.4167
*RcbHLH54*	*RcbHLH111*	0.438746	NaN	NaN	552	122.75	429.25
*RcbHLH55*	*RcbHLH106*	0.435383	1.727525	0.252027	681	142.3333	538.6667
*RcbHLH58*	*RcbHLH105*	0.367022	2.17115	0.169045	981	223	758
*RcbHLH60*	*RcbHLH102*	0.205017	1.25652	0.163163	753	174.1667	578.8333
*RcbHLH62*	*RcbHLH73*	0.230957	1.156031	0.199784	693	159.3333	533.6667
*RcbHLH64*	*RcbHLH72*	0.369998	1.720924	0.215	1137	258.75	878.25

**Table 3 ijms-24-16305-t003:** Plant *bHLH* family genes involved in disease resistance.

Gene Name	Gene ID	Species	Pathogens	References
*SlybHLH131*	Solyc06g051550.2.1	*Solanum lycopersicum*	*Tomato yellow leaf curl virus*	[13]
*FAMA*	AT3G24140	*Arabidopsis thaliana*	*Botrytis cinerea*	[22]
*AtMYC2*	At1g32640	*Arabidopsis thaliana*	*Botrytis cinerea*	[23]
*AtbHLH13*	At1g01260	*Arabidopsis thaliana*	*Botrytis cinerea*	[24]
*OsbHLH6*	Os04g23550	*Oryza sativa*	*Magnaporthe oryzae*	[25]
*OsbHLH034*	Os02g49480	*Oryza sativa*	*Xanthomonas oryzae pv. oryzae*	[14]

**Table 4 ijms-24-16305-t004:** Expression patterns of *RcbHLH* genes under infection of *B. cinerea*.

Gene ^2^	Accession Number	Group	log_2_Ratio 30 hpi	log_2_Ratio 48 hpi
*RcbHLH4*	RchiOBHm_Chr1g0348781	Ⅶ	−1.02302	−1.36247
*RcbHLH8*	RchiOBHm_Chr1g0361191	Ⅰa	−1.05954	−1.91491
*RcbHLH16*	RchiOBHm_Chr2g0091241	Ⅹ	0	−1.13271
** *RcbHLH17* **	**RchiOBHm_Chr2g0093571**	**Ⅳd**	**3.07535**	**4.92649**
** *RcbHLH21* **	**RchiOBHm_Chr2g0105931**	**Ⅶ**	**0**	**1.03517**
***RcbHLH29*** *****	**RchiOBHm_Chr2g0126861**	**Ⅰb**	**0**	**6.04668**
*RcbHLH32*	RchiOBHm_Chr2g0152511	Ⅶ	0	−16.01
** *RcbHLH34* **	**RchiOBHm_Chr2g0176421**	**Ⅲ(a+b+c)**	**1.07031**	**1.74663**
*RcbHLH37*	RchiOBHm_Chr3g0457291	ⅩⅥ	−1.36188	--
*RcbHLH39*	RchiOBHm_Chr3g0462431	Ⅷ(a+b+c)	−1.48345	--
** *RcbHLH40* **	**RchiOBHm_Chr3g0465361**	**ⅩⅢ**	**1.40229**	**2.20545**
*RcbHLH42*	RchiOBHm_Chr3g0480751	Ⅻ	0	−1.48859
** *RcbHLH44* **	**RchiOBHm_Chr4g0390311**	**Ⅳd**	**2.8578**	**4.76511**
** *RcbHLH46* **	**RchiOBHm_Chr4g0399211**	**Ⅲ(a+b+c)**	**1.25346**	**3.44205**
*RcbHLH50*	RchiOBHm_Chr4g0412071	Ⅻ	−1.46896	−1.77088
*RcbHLH53*	RchiOBHm_Chr4g0425781	ⅩⅥ	0	−1.61078
*RcbHLH55*	RchiOBHm_Chr4g0434901	Ⅻ	0	−2.09156
*RcbHLH57*	RchiOBHm_Chr4g0437041	Ⅻ	1.04454	--
** *RcbHLH59* **	**RchiOBHm_Chr4g0443741**	**Ⅲ(a+b+c)**	**0**	**1.92286**
*RcbHLH60* *	RchiOBHm_Chr4g0445091	Ⅻ	−1.22975	−4.12905
** *RcbHLH62* **	**RchiOBHm_Chr5g0004471**	**Ⅳa**	**0**	**2.03271**
** *RcbHLH67* **	**RchiOBHm_Chr5g0010631**	**Ⅴb**	**1.17478**	**1.30658**
** *RcbHLH72* **	**RchiOBHm_Chr5g0036871**	**Ⅹ**	**0**	**2.48493**
** *RcbHLH75* **	**RchiOBHm_Chr5g0053301**	**Ⅴb**	**−2.3709**	**−1.67965**
*RcbHLH78*	RchiOBHm_Chr5g0077341	Ⅸ	−1.05564	−1.12357
***RcbHLH80*** *****	**RchiOBHm_Chr6g0246251**	**Ⅳa**	**−3.29702**	**−2.05486**
*RcbHLH84*	RchiOBHm_Chr6g0264701	Ⅻ	0	−1.21958
** *RcbHLH90* **	**RchiOBHm_Chr6g0283511**	**Ⅻ**	**1.64315**	**2.09937**
*RcbHLH91*	RchiOBHm_Chr6g0285491	Ⅶ	0	−1.17746
*RcbHLH92*	RchiOBHm_Chr6g0288541	ⅩⅢ	0	−1.37089
*RcbHLH94*	RchiOBHm_Chr6g0289601	Ⅺ	0	−1.06842
** *RcbHLH99* **	**RchiOBHm_Chr6g0308251**	**Ⅶ**	**1.31529**	**2.89317**
** *RcbHLH101* **	**RchiOBHm_Chr6g0310101**	**Ⅻ**	**0**	**1.23509**
*RcbHLH102*	RchiOBHm_Chr7g0180121	Ⅻ	0	−1.80483
*RcbHLH104*	RchiOBHm_Chr7g0182341	Ⅶ	0	−1.28473
*RcbHLH105*	RchiOBHm_Chr7g0183781	Ⅴa	−1.38441	--
** *RcbHLH106* **	**RchiOBHm_Chr7g0185551**	**Ⅻ**	**0**	**1.81513**
** *RcbHLH108* **	**RchiOBHm_Chr7g0187141**	**Ⅲ(d+e+f)**	**0**	**2.74536**
*RcbHLH109*	RchiOBHm_Chr7g0187261	Ⅲ(d+e+f)	−3.56492	--
*RcbHLH110*	RchiOBHm_Chr7g0188921	Ⅲ(a+b+c)	0	−1.77728
** *RcbHLH111* **	**RchiOBHm_Chr7g0189021**	**Ⅻ**	**1.34134**	**2.0827**
***RcbHLH112*** *****	**RchiOBHm_Chr7g0193761**	**Ⅰb**	**1.22723**	**4.82187**
** *RcbHLH115* **	**RchiOBHm_Chr7g0209751**	**Ⅲ(d+e+f)**	**0**	**1.34433**
*RcbHLH116*	RchiOBHm_Chr7g0210101	Ⅰa	0	−3.38838
*RcbHLH118*	RchiOBHm_Chr7g0227911	Ⅸ	0	−1.0994
*RcbHLH121*	RchiOBHm_Chr7g0237511	ⅩⅢ	0	−1.02865

The log2 transformed expression profiles were obtained from the RNA-seq dataset [20]. ^2^ RcbHLHs upregulated are shown in bold. The genes validated by qPCR were marked with asterisks.

**Table 5 ijms-24-16305-t005:** List of primers used in this study.

Gene Name	Accession Number	Primer Sequence (5′-3′)	Amplicon Length	Ta	Tm	Amplification Efficiency
*RcbHLH29*	RchiOBHm_Chr2g0126861	F: GGTTCCACCCTAGAGGTTGTT	110 bp	60 °C	81.69	2.005
		R: CTGCACGGACTAGGTGAAGT				
*RcbHLH60*	RchiOBHm_Chr4g0445091	F: CGATGAGTTTGGACCACCGA	116 bp	60 °C	84.1	1.972
		R: CCTCAGCTTTGGCCTCAAGA				
*RcbHLH80*	RchiOBHm_Chr6g0246251	F: ACACAAACCAAGTGGGGGTT	102 bp	60 °C	85.27	1.968
		R: GTTCCCTGACTGGCCTTCAA				
*RcbHLH112*	RchiOBHm_Chr7g0193761	F: CGATCTTGCAGCCTCCTACA	120 bp	60 °C	82.43	2.024
		R: CAACCTTGATCCGACCACCA				
*RcUBI2*	RchiOBHm_Chr1g0359561	F: GCCCTGGTGCGTTCCCAACTG	82 bp	60 °C	82.43	2.024
		R: CCTGCGTGTCTGTCCGCATTG				

Ta: amplification temperature; Tm: melting temperature.

## Data Availability

The datasets used and/or analysed during the current study has been included within supplemental data. The plant materials are available from the corresponding author on reasonable request.

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
