# Peer review of "The Basic/Helix-Loop-Helix Transcription Factor Family Gene RcbHLH112 Is a Susceptibility Gene in Gray Mould Resistance of Rose (Rosa Chinensis)"

_ijms, 2023, doi:10.3390/ijms242216305_

Round 1

Reviewer 1 Report

Comments and Suggestions for Authors

Congratulations for this work which was explained in your manuscript in a simple and clear way. It is a nice approach from a scientific perspective to a big issue in flower industry. Your work applied bioinformatic tools to make good use of the available sequencing data and databases. You also made on-bench validation and applied state of the art techniques like VIGS to bring new knowledge about the function of a newly identified transcription factor in the infection of Botrytis cinerea in such important crop as rose. Please note that my comments are aiming to improve the manuscript in robustness.

Format/writing issues:

When an abbreviation appears in the text for the first time, you should write before that abbreviation what it stands for. In addition, not all the abbreviations used in the text are listed in the list of abbreviations (lines 336-342). In this fashion, I detected some examples that should be addressed:

- Line 12: The basic/helix-loop-helix family (bHLH) family...

- Line 20: Ratios of non-synonymous to synonymous mutation frequencies (Ka/Ks)...

- Line 106: ..all of which were whole genome duplication (WGD) or segmental duplication...

- Line 122: "We used the Neighbor-joining method (NJ)..."

- Line 202: Tobacco rattle virus (TRV2) vector

Writing:

- line 49: ..transcription factors are involved (not is involved)

- Line 24: "...is a negative regulator of rose to B. cinerea." According to your results, the genes is not "a regulator of the pathogen" even less "negative" to B. cinerea. Instead of this, it should be written in the abstract what is written in the results (line 209-210) "..is a suceptibility factor for rose towards the infection with B. cinerea"

- Lines 125-127: Figure 3 should be mention the first time is mentioned. Instead of being in line 127 should be in line 125, right after "phylogenetic analysis"

-Lines 150-167: The references mentioned in table 3 shouldn't be below table 3. They should be part of the list of references at the end of the paper. If it is more handy to codify them as numbers so they can fit in the last column of table 3, maybe you can change all citations to numbers, If it is not possible because the format of the journal, maybe you can add them as a separate section at the end of the list of references.

- Lines 169-172 (figure 4 legend): " The Composite phylogenetic tree  of that included all rose and Arabidopsis bHLH transcription factors and the bHLH transcription factors from Arabidopsis (Arabidopsis thaliana) were constructed by Neighbor-Joining method. The bootstrap...

- Line 173: "The Expression of.."

- Line 182: "RcbHLHs up-regulated that induce upregulated expression are shown in bold"

- Line 224: "And Wwe demostrated that.."

- Line 232: You stated that the RcbHLH 231 gene family is usually under purifying selection. What "usually" means in this context? Please elaborate or remove "usually".

- Line 249: The article "the" has to disappear "it has the negative regulation...". Indeed I would recommend to change that phrase by something like "...it is a susceptibility factor of rose in B. cinerea infection process." I am suggesting this change because your research clearly concludes that this transcriptional factor is a susceptibility factor. There is no use in mentioning the word "resistance" here. Instead, You could argue in the discussion or conclusions about the fact that you could create genome edited plants in the future more resistant to B. cinerea by knocking down RcbHLH112 gene.

- Line 288: ", as mentioned above, .." RNA extraction was not mentioned above nor in the entire manuscript. Therefore, that expression should be deleted.

- Line 306: It says that a minimum of 48 petals were use for each gene and VIGS were repeated, but in this work you only tested one gene and 2 treatments (empty vector and TRV-RcbHLH112).

- Line 318: Consistently with which I advised for line 249, I would say that the genes is involved in the susceptibility of rose to gray mould disease, not in the resistance. "Regulation of resistance" is only misleading and looks like an attempt to attract google searches for grey mould resistance.

Quality of figures:

-Fig 1, fig 3 and fig 4 have insufficient resolution in the pdf given. I can barely read anything, not even zooming. Resolution or size should be improved.

- Fig 5 (only suggestion): I would recommend to show the relative expression of the genes compared only to internal control (RhUbi) instead of showing the expression at 30hpi or 48hpi in relation to control expression (30h-PDB or 48h-PDB). This way each histogram would include 4 bars instead of 2 bars, but the reader could compare the expression of RcbHLH112 with the expression of control petals in figure 6C, for which I also recommend to show the expression relative to RhUbi instead to relative to the control itself (that is why the expression is 1 in the control in fig 6). If you decide to keep fig 5 that way, at least you should clarify what it is 30h/30h-PDB in the figure legend.

Missing information:

- Lines 79-82: The paragraph leads to confusion. Looks like MEME and Pfam db comparison confirmed all 136 candidates but them there is no explanation why only 121 were identified. I guess that from those 136 candidates detected by Hmmsearch with PF00010 as Query, only 121 protein sequences had a domains profile consistent with a bHLH gene. One suggestion is to change "... and Pfam database comparison further confirmed wether the extracted protein domain profile was consistent with the characteristics of the family...". This should be also clarified in materials and methods (line 259) "...of the above 136 rose bHLH candidates were extracted....to determine the final 121 family members".

- There is no mat and meth section for the mycrosyntenic analysis showed in Figure 2. Only the leyed mentions the software Circos and no reference is cited about this.

- Table 4: The chosen genes for qPCR validation should be indicated in the table so is easier for the reader to compare the RNASeq expression data to the qPCR results shown in figure 5.

-Line 189: Why those specific 4 genes were chosen for validation? Were they more interesting for some reason? Was because the difference of expression between 30hpi and 48hpi was bigger? why only 4? As it is written right now, it might lead to think that they were the only 4 genes that showed the same differential expression as the RNASeq...

- Lines 204-206: Explain better either her or in mat and meth how the control rose petals are treated. If the pTRV2 vector is empty why is called TRV-GFP? Does it contains green fluorescence protein? In that case, does TRV-RcbHLH112 contains GFP too? Is it the control also agroinfiltrated with VRT1 strain? It should also be clear in these lines that you inoculated the petals 6 days post-agroinfiltration and that the expression of RcbHLH112 was measured then (at 0hpi). This information is not clear in figure 6 nor in its legend.

Line 218: It should be clear that at 6 days post-silencing is right before the infection with B. cinerea (0hpi).

Lines 294-295 (suggestion): I believe that a little table of 5 primer pairs shouldn't take that much space. I think you could move Suppl Table S1 to the main text as Table 5 or in the form of text as you did in lines 298-300.

Methodological defects:

- The results shown in figure 5 (qPCR) are based in 3 technical replicates, not biological replicates. I assume the 3 technical replicates are 3 PCR reactions of the same cDNA? Normally is accepted at least 2 biological replicates (2 infection experiments) that implies 2 independent RNA extraction, cDNA synthesis and 3 technical replicates for each... This is not clear in results nor y material and methods. Indeed, there is no section in mat and meth for the inoculation of B. cinerea for the expression analysis. It is only explained for the section of VIGS. Did you use the control petal discs of the phenotyping of the VIGS experiment to do the expression analysis? In any case it should be clarified the number of experiments and samples used for qRT-PCR and if it is only one, it should be repeated.

- The relative expression post-silencing was only measured right before inoculating the petals (fig 6 C). It is not clear whether the expression was normalized in relation to RhUbi in this case as in fig 5. It looks like for both samples (TRV-GFP and TRV-RcbHLH112) the expression was divided by the expression of TRV-GFP. Table 4 and fig 5 clearly show that the expression of RcbHLH112 is up-regulated during infection with B. cinerea at 30hpi and 48hpi. I wonder why you didn't measure its expression in RV-GFP and TRV-RcbHLH112 petals 60hpi with B. cinerea as the samples phenotyped in fig 6A and B, or at 30hpi or 48hpi or even earlier stages of B. cinerea infection to test the efficiency of VIGS silencing in petals where the expression should be really high. That result could help in the discussion; Maybe you could have got higher reductions in lesion size depending of the efficiency of silencing during the infection process. 

Author Response

Congratulations for this work which was explained in your manuscript in a simple and clear way. It is a nice approach from a scientific perspective to a big issue in flower industry. Your work applied bioinformatic tools to make good use of the available sequencing data and databases. You also made on-bench validation and applied state of the art techniques like VIGS to bring new knowledge about the function of a newly identified transcription factor in the infection of Botrytis cinerea in such important crop as rose. Please note that my comments are aiming to improve the manuscript in robustness.

>>Response: Thank you for giving us the opportunity to revise our article entitled " The Basic/Helix-Loop-Helix Transcription Factor Family gene RcbHLH112 is a susceptibility gene in gray mould re-sistance of rose (Rosa chinensis)". We appreciate the insightful comments and suggestions provided by you, which helped us improve the quality of our manuscript. Herein, we have addressed each comment raised and explained the changes made to the manuscript. Our responses to your comments are noted below (marked with ‘>>Response:’). In the revised manuscript, the yellow highlighted text indicates where changes have been made.

Format/writing issues:

When an abbreviation appears in the text for the first time, you should write before that abbreviation what it stands for. In addition, not all the abbreviations used in the text are listed in the list of abbreviations (lines 336-342). In this fashion, I detected some examples that should be addressed:

- Line 12: The basic/helix-loop-helix family (bHLH) family...

- Line 20: Ratios of non-synonymous to synonymous mutation frequencies (Ka/Ks)...

- Line 106: ..all of which were whole genome duplication (WGD) or segmental duplication...

- Line 122: "We used the Neighbor-joining method (NJ)..."

- Line 202: Tobacco rattle virus (TRV2) vector

>>Response: Thank you for your valuable feedback. We have addressed the issue of writing out the full form of abbreviations when they first appear in the text, as well as including all abbreviations in the list provided.

Writing:

- line 49: ..transcription factors are involved (not is involved)

- Line 24: "...is a negative regulator of rose to B. cinerea." According to your results, the genes is not "a regulator of the pathogen" even less "negative" to B. cinerea. Instead of this, it should be written in the abstract what is written in the results (line 209-210) "..is a suceptibility factor for rose towards the infection with B. cinerea"

- Lines 125-127: Figure 3 should be mention the first time is mentioned. Instead of being in line 127 should be in line 125, right after "phylogenetic analysis"

-Lines 150-167: The references mentioned in table 3 shouldn't be below table 3. They should be part of the list of references at the end of the paper. If it is more handy to codify them as numbers so they can fit in the last column of table 3, maybe you can change all citations to numbers, If it is not possible because the format of the journal, maybe you can add them as a separate section at the end of the list of references.

- Lines 169-172 (figure 4 legend): " The Composite phylogenetic tree  of that included all rose and Arabidopsis bHLH transcription factors and the bHLH transcription factors from Arabidopsis (Arabidopsis thaliana) were constructed by Neighbor-Joining method. The bootstrap...

- Line 173: "The Expression of.."

- Line 182: "RcbHLHs up-regulated that induce upregulated expression are shown in bold"

- Line 224: "And Wwe demostrated that.."

- Line 232: You stated that the RcbHLH 231 gene family is usually under purifying selection. What "usually" means in this context? Please elaborate or remove "usually".

- Line 249: The article "the" has to disappear "it has the negative regulation...". Indeed I would recommend to change that phrase by something like "...it is a susceptibility factor of rose in B. cinerea infection process." I am suggesting this change because your research clearly concludes that this transcriptional factor is a susceptibility factor. There is no use in mentioning the word "resistance" here. Instead, You could argue in the discussion or conclusions about the fact that you could create genome edited plants in the future more resistant to B. cinerea by knocking down RcbHLH112 gene.

- Line 288: ", as mentioned above, .." RNA extraction was not mentioned above nor in the entire manuscript. Therefore, that expression should be deleted.

- Line 306: It says that a minimum of 48 petals were use for each gene and VIGS were repeated, but in this work you only tested one gene and 2 treatments (empty vector and TRV-RcbHLH112).

- Line 318: Consistently with which I advised for line 249, I would say that the genes is involved in the susceptibility of rose to gray mould disease, not in the resistance. "Regulation of resistance" is only misleading and looks like an attempt to attract google searches for grey mould resistance.

>>Response: Thank you for your carefully reading and noticed these issues. We have addressed all these issue of writing.

Quality of figures:

-Fig 1, fig 3 and fig 4 have insufficient resolution in the pdf given. I can barely read anything, not even zooming. Resolution or size should be improved.

- Fig 5 (only suggestion): I would recommend to show the relative expression of the genes compared only to internal control (RhUbi) instead of showing the expression at 30hpi or 48hpi in relation to control expression (30h-PDB or 48h-PDB). This way each histogram would include 4 bars instead of 2 bars, but the reader could compare the expression of RcbHLH112 with the expression of control petals in figure 6C, for which I also recommend to show the expression relative to RhUbi instead to relative to the control itself (that is why the expression is 1 in the control in fig 6). If you decide to keep fig 5 that way, at least you should clarify what it is 30h/30h-PDB in the figure legend.

>>Response: Thank you very much for your review and comments on the quality of the figures in our paper. We greatly appreciate your suggestions and fully agree that the readability and clarity of the figures are crucial.

Firstly, we completely understand and agree with your concerns regarding the low resolution of Fig 1, Fig 3, and Fig 4. In fact, the original images we have are high-resolution, but the resolution was compromised during the conversion to PDF format. We will provide the journal with the high-resolution original images so that readers can directly download them, and we will request the journal to take measures to improve the resolution or size of the images in the PDF file to ensure clear readability.

Furthermore, we deeply appreciate your suggestion regarding Fig 5. Allow me to explain the specific calculation process. Essentially, we first calculate the expression level of the bHLH gene relative to the housekeeping gene (RhUbi) under Botrytis infection conditions (30h). Simultaneously, we calculate the expression level of the bHLH gene relative to the housekeeping gene (RhUbi) under mock infection conditions (30h-PDB). Finally, we compare the values of 30h and 30h-PDB to obtain reliable conclusions (30h/30h-PDB). We believe that this approach provides a clearer representation of the effect of Botrytis on bHLH and makes it easier for readers to understand. Therefore, we have decided to retain the format of Fig 5. Additionally, we accept your suggestion and will clearly explain the meaning of 30h/30h-PDB in the figure legend to eliminate any possible ambiguity.

Missing information:

- Lines 79-82: The paragraph leads to confusion. Looks like MEME and Pfam db comparison confirmed all 136 candidates but them there is no explanation why only 121 were identified. I guess that from those 136 candidates detected by Hmmsearch with PF00010 as Query, only 121 protein sequences had a domains profile consistent with a bHLH gene. One suggestion is to change "... and Pfam database comparison further confirmed wether the extracted protein domain profile was consistent with the characteristics of the family...". This should be also clarified in materials and methods (line 259) "...of the above 136 rose bHLH candidates were extracted....to determine the final 121 family members".

>>Response: We apologize for the confusion caused by the lack of explanation regarding the identification of 121 out of the initially confirmed 136 candidates. Your understanding is correct; only 121 protein sequences had a domain profile consistent with a bHLH gene. We will revise the paragraph in question to explicitly state this, as well as update the Materials and Methods section (line 259) to clearly explain that the final 121 family members were determined based on the consistency of their extracted protein domain profiles.

- There is no mat and meth section for the mycrosyntenic analysis showed in Figure 2. Only the leyed mentions the software Circos and no reference is cited about this.

>>Response: We used TBtools to analyse the collinearity/microsyntenic of bHLH members . We included this in the Materials and Methods section, with a refence (Wang et al., 2012). Additionally, we will provide a reference citation for the use of Circos in our analysis.

- Table 4: The chosen genes for qPCR validation should be indicated in the table so is easier for the reader to compare the RNASeq expression data to the qPCR results shown in figure 5.

>>Response: We appreciate your suggestion to indicate the chosen genes for qPCR validation in the table. This will facilitate easier comparison between the RNASeq expression data and the qPCR results shown in Figure 5. We will update the table accordingly to provide this information.

-Line 189: Why those specific 4 genes were chosen for validation? Were they more interesting for some reason? Was because the difference of expression between 30hpi and 48hpi was bigger? why only 4? As it is written right now, it might lead to think that they were the only 4 genes that showed the same differential expression as the RNASeq...

>>Response: We acknowledge your concern regarding the selection of the specific four genes for validation. In fact, they were chosen at random. We will clarify this point to avoid any ambiguity.

- Lines 204-206: Explain better either her or in mat and meth how the control rose petals are treated. If the pTRV2 vector is empty why is called TRV-GFP? Does it contains green fluorescence protein? In that case, does TRV-RcbHLH112 contains GFP too? Is it the control also agroinfiltrated with VRT1 strain? It should also be clear in these lines that you inoculated the petals 6 days post-agroinfiltration and that the expression of RcbHLH112 was measured then (at 0hpi). This information is not clear in figure 6 nor in its legend.

>>Response: We apologize for the lack of clarity regarding the treatment of control rose petals. The pTRV2 vector does contain green fluorescent protein (GFP). In fact our lab usually uses TRV with GFP sequences as controls, which is also a common practice in many labs. We will revise the text to clearly state that TRV with GFP is used as a control. Furthermore, we will include in these lines that the petals were inoculated 6 days post-agroinfiltration, and the expression of RcbHLH112 was measured at 0hpi, providing additional clarity to the experimental timeline.

Line 218: It should be clear that at 6 days post-silencing is right before the infection with B. cinerea (0hpi).

>>Response: Thank you. We will make it explicit in the manuscript to avoid any confusion.

Lines 294-295 (suggestion): I believe that a little table of 5 primer pairs shouldn't take that much space. I think you could move Suppl Table S1 to the main text as Table 5 or in the form of text as you did in lines 298-300.

>>Response: We appreciate your suggestion to include a table for the primer pairs. We agree that it can be accommodated within the main text without significant space constraints. As such, we will move Supplementary Table S1 to the main text as Table 5.

Methodological defects:

- The results shown in figure 5 (qPCR) are based in 3 technical replicates, not biological replicates. I assume the 3 technical replicates are 3 PCR reactions of the same cDNA? Normally is accepted at least 2 biological replicates (2 infection experiments) that implies 2 independent RNA extraction, cDNA synthesis and 3 technical replicates for each... This is not clear in results nor y material and methods. Indeed, there is no section in mat and meth for the inoculation of B. cinerea for the expression analysis. It is only explained for the section of VIGS. Did you use the control petal discs of the phenotyping of the VIGS experiment to do the expression analysis? In any case it should be clarified the number of experiments and samples used for qRT-PCR and if it is only one, it should be repeated.

>>Response: Thank you very much for your review comments. We apologize for any lack of clarity regarding the VIGS and qPCR. In fact, VIGS has been repeated for three times, and our conclusions were based on the results of these three repeats. At the same time, qPCR was not performed with three PCR reactions of the same cDNA, the cDNAs are from different petals. We will make the necessary revisions in the revised manuscript, to make this more clear.

- The relative expression post-silencing was only measured right before inoculating the petals (fig 6 C). It is not clear whether the expression was normalized in relation to RhUbi in this case as in fig 5. It looks like for both samples (TRV-GFP and TRV-RcbHLH112) the expression was divided by the expression of TRV-GFP. Table 4 and fig 5 clearly show that the expression of RcbHLH112 is up-regulated during infection with B. cinerea at 30hpi and 48hpi. I wonder why you didn't measure its expression in RV-GFP and TRV-RcbHLH112 petals 60hpi with B. cinerea as the samples phenotyped in fig 6A and B, or at 30hpi or 48hpi or even earlier stages of B. cinerea infection to test the efficiency of VIGS silencing in petals where the expression should be really high. That result could help in the discussion; Maybe you could have got higher reductions in lesion size depending of the efficiency of silencing during the infection process. 

>>Response: Thank you for your valuable feedback regarding our manuscript.

Regarding the relative expression post-silencing measurement in Fig. 6C, we apologize for not clearly indicating whether the expression was normalized in relation to RhUbi or not. We confirm that the normalization was performed in the same manner as in Fig. 5, where the expression of RcbHLH112 was normalized to the expression of RhUbi. We will include a clear statement in the figure legend to avoid any confusion.

Regarding measuring the expression of RcbHLH112 in TRV-GFP and TRV-RcbHLH112 petals at different time points, we appreciate your suggestion. We only measured the expression before inoculating the petals as we were interested in determining the efficiency of silencing at this stage. Inoculating and then testing for silencing efficiency would complicate the conditions of the experiment to the detriment of drawing clear conclusions.

Reviewer 2 Report

Comments and Suggestions for Authors

The paper is well-structured and maintains a clear focus, with the title effectively conveying the paper's subject. The study identified and categorized 121 RcbHLH genes in the rose genome into 21 sub-groups, highlighting their diversity and distribution across all seven chromosomes.

The introduction effectively introduces the significance of transcription factors, particularly the bHLH gene family, in plant biology. It traces the historical development of this field and explains the multifunctional roles of bHLH genes in plants. Though is short can be improved with citing similar transcription factor.  

Authers identified 121 RcbHLH genes in the rose genome through a series of bioinformatics analyses. These genes were unevenly distributed across seven chromosomes, with chromosome 6 having the highest number. Tandem and segmental duplications were identified in 16 gene pairs, indicating gene expansion. Phylogenetic analysis revealed the evolutionary relationships among these genes. These findings provide insights into the diversity and evolution of the bHLH gene family in roses. Figure presetation need improviment quality is not good, difficult to understand.

The discussion offers insightful interpretations of the results, particularly concerning gene duplication events and selection pressure, but could benefit from expanding on the functional implications of RcbHLH genes for grey mold resistance in roses.

The methods section is thorough and appropriately utilizes various analyses to characterize RcbHLH genes. The results are presented in a well-organized manner, with tables aiding in the presentation of gene-specific data. However, incorporating visual aids, such as figures or diagrams, could enhance the clarity of complex analyses.

Comments on the Quality of English Language

Ensure consistent terminology throughout the manuscript. For example, the use of "bHLH" and "bHLHs" could be standardized for clarity. Some sentences are quite long and complex. Consider breaking them into smaller sentences to improve readability.

Author Response

The paper is well-structured and maintains a clear focus, with the title effectively conveying the paper's subject. The study identified and categorized 121 RcbHLH genes in the rose genome into 21 sub-groups, highlighting their diversity and distribution across all seven chromosomes.

>>Response: Thank you for your detailed and constructive feedback on our manuscript. We appreciate your positive comments regarding the structure and focus of the paper, as well as the effectiveness of the title in conveying the subject matter. We are glad to hear that you found our study on the identification and categorization of RcbHLH genes in the rose genome informative, highlighting their diversity and distribution across chromosomes. Our responses to your comments are noted below (marked with ‘>>Response:’). In the revised manuscript, the yellow highlighted text indicates where changes have been made.

The introduction effectively introduces the significance of transcription factors, particularly the bHLH gene family, in plant biology. It traces the historical development of this field and explains the multifunctional roles of bHLH genes in plants. Though is short can be improved with citing similar transcription factor. 

Authers identified 121 RcbHLH genes in the rose genome through a series of bioinformatics analyses. These genes were unevenly distributed across seven chromosomes, with chromosome 6 having the highest number. Tandem and segmental duplications were identified in 16 gene pairs, indicating gene expansion. Phylogenetic analysis revealed the evolutionary relationships among these genes. These findings provide insights into the diversity and evolution of the bHLH gene family in roses. Figure presetation need improviment quality is not good, difficult to understand.

The discussion offers insightful interpretations of the results, particularly concerning gene duplication events and selection pressure, but could benefit from expanding on the functional implications of RcbHLH genes for grey mold resistance in roses.

The methods section is thorough and appropriately utilizes various analyses to characterize RcbHLH genes. The results are presented in a well-organized manner, with tables aiding in the presentation of gene-specific data. However, incorporating visual aids, such as figures or diagrams, could enhance the clarity of complex analyses.

>>Response: Firstly, we acknowledge your suggestion to improve the introduction by citing similar transcription factor studies. We revised the introduction accordingly to provide a more comprehensive overview of the existing literature and emphasize the relevance of our research within the broader context of transcription factors in plant biology.

Secondly, we appreciate your insightful comment on expanding the discussion to include the functional implications of RcbHLH genes for grey mold resistance in roses. Studying the mode of action of this susceptibility gene, RcbHLH112, is one of the current focuses of research in our lab, but we can't give more clues in the discussion as far as the current results are concerned. But I'm sure in the future we can further elaborate on this aspect to provide a more comprehensive understanding of the potential role of RcbHLH genes in disease resistance mechanisms.

Finally, regarding the quality of figures, we completely understand your concerns. In fact, the original images we made are high-resolution, but the resolution was compromised during the conversion to PDF format. We will provide the journal with the high-resolution original images so that readers can directly download them, and we will request the journal to take measures to improve the resolution or size of the images in the PDF file to ensure clear readability.